# Climate Change Adaptation and the Agriculture–Food System in Myanmar

Aung Tun Oo [1,*], Duncan Boughton [2] and Nilar Aung [2]

1   Radanar Ayar Rural Development Association, Bogale 10231, Myanmar
2   Department of Agricultural, Food and Resource Economics, Michigan State University,
    East Lansing, MI 48824, USA; boughton@msu.edu (D.B.); aungnila@msu.edu (N.A.)
*   Correspondence: aunghtunoo717.yau@gmail.com; Tel.: +95-9-453-711922

**Abstract:** The agricultural sector provides employment and income to the majority of Myanmar's population. The sector, however, is extremely susceptible to severe weather, rising temperatures, and changes in precipitation. A lack of knowledge about farming communities' climate change vulnerabilities, as well as the insufficient integration of policies and programs, is a constraint to climate change adaptation in agriculture sectors. This paper analyzes the drivers of the agricultural sector's vulnerability to climate change and highlights the key production systems that are most at risk in Myanmar. The paper examines historical climate information and the anticipated effects of climate change. We include an in-depth literature review and summaries of climate change adaptation efforts in agriculture sectors, along with recommendations for targeted, locally appropriate actions to strengthen the resilience of the agricultural sector. Farm households use a combination of scientific and indigenous adaptation strategies to cope with the effects of climate change. Additionally, the study reviews Myanmar's institutional framework for climate action and government priorities for adaptation measures, emphasizes the urgent need for climate action in agriculture sectors, and calls for more research and development efforts on the effects of climate change on rural livelihoods and agriculture.

**Keywords:** climate change; agriculture; Myanmar; adaptation measures; resilience

## 1. Introduction

Globally, climate change is now widely recognized as one of the most difficult issues of the twenty-first century [1]. The frequency and severity of extreme climate conditions such as droughts and floods in agriculturally sensitive regions have increased as temperatures have risen and precipitation patterns have shifted [1,2]. Climate impact assessments also anticipate regional differences in agricultural productivity. While tropical zones are expected to experience negative effects on food supply, high-latitude regions are expected to experience some positive effects. Global warming has the potential to reduce crop yields and food security globally, necessitating the proper management of climate change adaptation and mitigation measures across all components of the agriculture sector [1,3].

According to the Global Climate Risk Index, Myanmar was ranked second in the degree of vulnerability to severe weather between 2000 and 2019 [2]. Myanmar is particularly vulnerable to extreme weather events, which have increased in frequency and intensity over the past 60 years [4,5]. Because its biological processes depend on hydro-climatic conditions, agriculture is particularly sensitive to extreme weather events, rising temperatures, and changes in precipitation [6–8].

Myanmar has an abundance of natural resources, and its 676,575 km$^2$ of land area includes a wide variety of topographic features and ecological zones. Due to its rich natural resources (land and water), the agriculture sector plays a key role in economic growth and income for the majority of the population. Nearly 60% of the labor force is employed in agriculture, livestock, and fishing sectors, which contributes to around 30% of GDP [9].

Myanmar's four major agroecological zones are the delta, coastal zone, dry zone, and highland areas. Agricultural output and population are concentrated predominantly in the delta and central dry zones [10,11]. The central dry zone (CDZ), which makes up about 60% of all croplands in Myanmar, is susceptible to extreme heat, water scarcity, and drought. Highland regions feature more tree and horticulture crops, as well as maize. The coastal regions to the south are regularly threatened by tropical cyclones, floods, and other storm-surge effects, which also pose a threat to the highly populated delta regions [4,7,12].

Due to climate variability across the country, Myanmar is confronted with numerous challenges and difficulties, including lowered crop yields and higher crop losses, food insecurity, malnutrition, and poverty [13]. To attain sustainable adaptation in the agriculture sector, it is critical to identify the impacts of climate change on agriculture and rural livelihoods, the main barriers to the adoption of climate change adaptation measures, how the institutional setting affects these barriers, and what the challenges are to integrating climate change adaptation policies to address agriculture–food system resilience. Through a systematic review of studies on climate change impacts, vulnerability, and adaptation measures in the agriculture sector, our paper provides more precise, regionally tailored recommendations for improving the resilience of the agriculture sector. This review will assist stakeholders from the government and non-government sectors, including researchers and policymakers, to address climate change threats in Myanmar from the perspective of agricultural adaptation.

## 2. Climate Change Impacts and Vulnerability

### 2.1. Historic Weather and Climate

According to historical data on weather and climate patterns, Myanmar is currently experiencing a period of increasing temperatures, more erratic rainfall, and a progressive alteration in monsoon patterns. The Department of Meteorology and Hydrology (DMH) of Myanmar claims that climate change has been noticeable over the past 60 years. Over three decades (1981–2010), the mean temperature has risen by around 0.08 °C per decade, with significant regional variations (see Supplementary Figures S1–S4). Inland regions experienced a greater average temperature increase (0.35 °C per decade) compared to coastal regions (0.14 °C per decade). In the central dry zone, extreme temperature changes (an increase of 2.4 °C on the baseline temperature data in 30 years), droughts, and water scarcity for people and livestock were observed.

The intensity and distribution of rainfall patterns have also changed (see Supplementary Figure S5). In recent years, the southwest monsoon has arrived later and departed earlier with heavier rainfall and harsher weather. Annual rainfall increased in the northern hilly region by 228 mm between 2001 and 2020 but fell in the Ayeyarwady, Tanintharyi, and Yangon regions, as well as Rakhine State, by 58 mm. Sea levels are rising in coastal areas, and there has also apparently been an increase in saltwater intrusion onto farmlands [4,7,14].

In Myanmar, since the year 2000, cyclones now occur almost annually compared to once every three years on average in preceding decades. The livelihoods and food security of farm households in the delta region and coastal regions of Myanmar are particularly at risk from cyclones and strong storms. Between 1968 and 2008, Myanmar was hit by seven severe cyclones. The worst was Cyclone Nargis in 2008, which destroyed almost 10 million acres of rice (57 percent of the total annual rice-producing areas), killed approximately 140,000 people, and resulted in USD 10 billion in losses [12,15]. Twelve significant floods affected the nation between 1910 and 2000, but the frequency has increased in recent decades [7]. In Myanmar, 12 out of 14 regions/states were affected by floods in 2015, which had devastating impacts on agriculture and resulted in 132 fatalities [4,8].

### 2.2. Projected Weather and Climate

According to recent climate change forecasts, extreme weather events will continue to occur more frequently in Myanmar, in terms of temperature, rainfall, and sea level rise (see Supplementary Figures S7–S9) [4,8]. In Myanmar, projected increases in precipitation

of 11% by 2040 and 23% by 2070, combined with a shorter monsoon season, may result in unexpected, heavy precipitation that could result in flash floods, riverine, and pluvial floods [16]. Floods are the most common average annual disaster in Myanmar, followed by other miscellaneous accidents and storms, among other things (see Supplementary Figure S6). According to the projections, annual average temperatures between 2011 and 2040 will be 0.7 to 1.1 °C warmer than they were between 1980 and 2005. Temperature increases of 1.1 °C by 2040 and 2.7 °C by 2070 are anticipated. After 2040, the central dry zone is expected to experience temperature increases of up to 3 °C [4,16,17].

Inland regions are expected to experience an increase in temperature of 0.3 to 0.4 °C. During the hot season, it may also increase by up to 3 °C in eastern and northern hilly regions [16]. Along with that, it is forecasted that the sea level will increase by 5 to 13 cm in 2020, 20 to 41 cm in 2050, and 37 to 83 cm in 2080. This will cause more saltwater intrusion farther inland [8,16]. Myanmar's agriculture sectors are likely to be impacted by the projected impacts of climate change, and significant investment in agricultural resilience to climate change is required.

## 3. Key Agriculture Sector Vulnerabilities

### 3.1. Agro-Ecological Features and Agri-Food System Vulnerability

Nearly 70% of Myanmar's population resides in rural areas and is primarily dependent on small- and medium-sized businesses in the agriculture, fishing, and aquaculture sectors. Since agriculture accounts for a large portion of Myanmar's economy, the country's food production and food security may be adversely affected by the unprecedented effects of climate change in Myanmar [7,12]. Climate change affects agriculture productivity through changes in agro-ecological conditions. The delta region is the key agroecological zone for rice production in Myanmar because of its rich alluvial soils, favorable climatic conditions, and ample water supply. In the southern coastal region (particularly Tanintharyi and Mon Regions), while rice farming is also popular, perennial crops such as rubber, palm oil, and tropical fruit trees dominate the farming landscape.

The central dry zone consists of the western and central parts of the Mandalay Region and the lower parts of the Sagaing and Magway Regions. While Magway Region is the fifth-largest rice-growing region, Sagaing Region is the third-largest region for rice production. In Sagaing, Mandalay, and Magway Regions, agriculture employs 70%, 80%, and 95% of the total cropland [18]. Upland farming of crops, including pulses, sesame, and cotton, is prevalent in the central dry zone, and there is extensive cropping everywhere [19]. The CDZ is considered one of the most climate-sensitive regions due to its increasing temperature, decreasing water availability, increased frequency of droughts, and severe weather effects [20,21].

Because rice is a staple food crop in Myanmar and a major export, it continues to dominate the agricultural sector. Rice production is dependent on precipitation and is concentrated in disaster-prone areas, especially the delta region and the upper parts of the central dry zone. This makes rice production particularly vulnerable to the effects of climate change, such as temperature changes, variations in rainfall, an increase in the frequency of delayed monsoon rains, a decrease in the duration of the monsoon season, and an increase in the severity of extreme weather events such as droughts, floods, and heat waves [4,5]. The perceptions of farmers on climate change variables and extreme events are presented in Table 1.

**Table 1.** Perception of farmers on climate change variables and extremes in Myanmar.

| Aspects | Climate and Weather Parameters | Perception of Farmers on Climate Change | Effects of Climate Change |
|---|---|---|---|
| Climate variables | -Temperature | Increase | Loss of production/income/foods |
| | -Precipitation | Increase with shorter monsoon/erratic nature | Loss of crops, properties, aquatic animals |
| | Hailstorm | Increase | Damage to farm crops and household properties |
| Natural hazards | -Tropical cyclones | Increase in frequency and intensity | Loss of crops, farm equipment, human lives, damage to farms, and draught animals |
| | -Flood | Increase in frequency and intensity | Crop damages and loss of farm equipment, household properties |
| | -Saltwater intrusion | Increase in frequency and intensity | Damages on paddy crops and certain fish species |
| Impacts on nature | -Agro-ecology | Increase | Change in agro-ecological conditions |
| | -Water resources (Ocean) | Increase | Alteration of ocean current |
| | -Soil | Increase in drought, crack in the ground | Water scarcity, drinking water deficit, groundwater depletion, crop damage |
| | -Pests and diseases | Increase in occurrence of pests and disease infestation | Loss of crops, high pesticides cost |
| | -Health and wellbeing | Increase in vulnerability of farm household's wellness | Absent from work/school, lower labor productivity |

Adapted from [8,16,22–25].

### 3.2. Water Resources

The Ayeyarwady, Chindwin, Sittaung, and Thanlwin Rivers flow from north to south in Myanmar and provide abundant water resources for crop production. Depending on seasonal fluctuations in water discharge and water surface level between wet and dry seasons, these principal rivers supply significant water resources with catchment areas of close to 737,000 km$^2$ [5]. About half of Myanmar, or 6 states out of 14 regions/states, are located in river basins where water is available all year [26,27]. Freshwater and marine waters, among others, are abundant in Myanmar. There are 8.2 million hectares of inland water bodies and 0.5 million hectares of interchange areas, respectively [20]. A considerable groundwater resource exists in Myanmar in addition to the availability of rivers and streams. As previously mentioned, groundwater cycles are also threatened by climate-change-related water availability, rising temperatures, precipitation, and saltwater intrusion (https://www.burmalibrary.org/sites/burmalibrary.org/files/obl/Myint-Thein-Climate-Change-Ground-water-2019-July-7-10-edited-en-red.pdf (accessed on 28 May 2023)). Precipitation patterns, particularly during the monsoon, determine the availability of water in the dry zone. Fresh water shortages are brought on by decreasing rainfall, which directly threatens groundwater replenishment, rivers, and water reservoirs.

The country's water supplies were reduced by the severe 2010 drought, which also influenced crop production across the country, most notably in the central dry zone [6,8,21]. However, during the past three decades, groundwater irrigation and river pumping systems have increased more quickly in the dry zone [19]. In addition, Myanmar has 132 dams across the country. Due to the deterioration of the watersheds, only 25% of these dams could function properly for either agriculture purposes or the generation of electricity. Recent investigations question the lack of access to water from irrigation projects despite the abundance of water resources and irrigation dams across the country [28–30].

## 4. Major Ecosystem Vulnerabilities

### 4.1. Upland

The geography of Myanmar is diverse, ranging from the lowland central dry zone and delta region to upland plateaus and hilly regions in the eastern, northern, and north-western regions. Table 2 shows the key vulnerable regions/states, major crops, and climate vulnerability status by agro-climatic zones. In upland plateaus and hilly areas, around 10% of the nation's total cultivated land is considered to be vulnerable to severe soil erosion. For example, Shan and Chin States, as well as the upper section of the Sagaing Region, are especially sensitive to land degradation and severe soil erosion due to the rolling terrain and high elevation. High deforestation, subpar farming methods, and shifting cultivation are just some contributing factors to soil erosion and land degradation in upland regions [4,5]. There are several different ways to grow rice, including terraced paddy in valleys, shifting rice farming, and lowland paddy culture at the bottom of small basins [31]. Fruits and vegetables, including tea leaves, avocados, cabbage, and cauliflower, as well as perennial crops such as coffee and avocado, are the main upland crops.

**Table 2.** Major agro-climate zones in Myanmar (source: authors).

| Agro-Climate Zones | Geographical Description | Vulnerable Regions and States | Major Crops/Livelihood Activities | Climate Hazards and Vulnerability Status [1] |
|---|---|---|---|---|
| Delta region | The Ayeyarwady River runs 1200 km (750 miles) from Upper Myanmar to the Andaman Sea. 50,400 km$^2$ of land areas 2500–5500 mm of annual rainfall [2] | Delta areas of Ayeyarwady and Yangon Riverine areas of the Bago region (i.e., Sittaung Riverine areas) | Rice and pulses | Cyclone, storm surges, intense rain, saltwater intrusion, Tsunami, and riverine flood Vulnerability status: high |
| Central dry zone | 87,198 km$^2$ or 12.8% of Myanmar's land area 500–1000 mm of annual rainfall [3] | Sagaing, Mandalay, and Magway Regions | Upland crops, oilseeds, pulses, rice, cotton, irrigated agriculture, and Kaing-Kyun (Silty land) cultivation | Drought, extremely high temperatures, flash floods, riverine floods, Deficit rainfall, and water scarcity Vulnerability status: extremely high |
| Coastal (upland and lowland) | 10–15% of the land area of Myanmar The average annual rainfall is highest in Myanmar at 3300 mm [4] | Tanintharyi, Mon Kayin, the Rakhine States, and some parts of the Ayeyarwady Region | Orchards, rice, pulses, upland agriculture, oilseeds, and nipa palm Fishing, fish-farming, fish processing | Cyclone/strong winds, Intense rain, sea level rise Vulnerability status: moderate to high |
| Hilly and mountainous areas | Hilly, uneven topography, sloping land, moderate to heavy rainfall | Shan, Kayin, Kachin, and Chin States Some parts of Kayin and Mon State | Upland crops, horticultural crops, and shifting cultivation | Intense rains, landslides Vulnerability status: low to moderate |

[1] Author evaluation of vulnerability indices based on existing literature and research findings; [2] [7]; [3] [22] [4] [22].

### 4.2. Lowland

The central dry zone of Myanmar is a huge valley that is 600 km long, 110 km wide, and covered in silt and clay-rich alluvial soils. These deep soil layers create a fertile area that is ideal for the growth of highland crops such as rice [8,32]. The CDZ climate is a hot, humid climate, while on the other hand, the delta region has a long coastline of about 2.400 km and is bordered to the south by the Bay of Bengal and the Andaman Sea. More than 95% of the crops produced in the region are rice [32]. Floods and droughts can have devastating effects on the lowlands and the central dry zone. The lowlands and flat terrains located in the river basins are particularly vulnerable to catastrophic flooding during the monsoon season [5]. Lowland regions of the country are mostly known for their high

production of rice, peas and beans, groundnuts, onions, lettuce, watermelons, bitter gourds, tomatoes, and various other fruit and vegetable crops [5,32].

## 5. National Strategies, Plans, and Institutions Relevant to Climate Change

In recent years, the government of Myanmar has developed several climate-related plans for the agricultural sector. To lessen the effects of natural disasters and climate change, various government departments have worked to introduce drought-resistant seed varieties, flood- and saltwater-tolerant varieties, and other cropping practices. The Myanmar National Adaptation Plan (NAPA), which was put into effect by the government in 2012, identifies eight priority sectors which are particularly vulnerable to climate change. These sectors include agriculture, early warning systems, forests, public health, water resources, the coastal lowlands, energy and industry, and biodiversity. To achieve food and nutrition security, the Myanmar Climate-Smart Agriculture Strategy (MCSA) was launched in 2015. Its goal is to increase agricultural productivity and climate resiliency by adopting a diversity of crop varieties and corresponding farming technology [22]. The MSDP (2018–2030) addresses the 2030 Sustainable Development Goals (SDGs), which call for mainstreaming climate action into all pertinent short-, medium-, and long-term national development plans and policies. The Myanmar Climate Change Policy (2019), Myanmar Climate Change Strategy (MCCS) (2018–2030), and Myanmar Climate Change Master Plan (MCCMP) (2018–2030) were all also adopted in 2019. Myanmar's comprehensive response to hazards associated with climate change is outlined in the Myanmar Climate Change Strategy and Action Plan (MCCSAP), a 15-year road map. The objective of the MCCASP is to generate and optimize opportunities for low-carbon and climate-resilient development in the nation, as well as to strengthen the adaptive capacity of vulnerable people and livelihood sectors [4,5].

### 5.1. Institutional Framework

To incorporate climate change into all pertinent short-, medium-, and long-term development plans and strategies, Myanmar has built an institutional framework. The Ministry of Natural Resources and Environmental Conservation (MONREC), the focal ministry, supported the establishment of the National Environmental Conservation and Climate Change Central Committee (NECCCCC) in 2016 to address and tackle environmental and climate change issues at the national/union level. Twenty representatives from 18 ministries, including the Ministry of Agriculture, Irrigation and Livestock (MOALI), one non-governmental organization (NGO), one private sector, and one civil society organization (CSO) make up the NECCCCC. There are also six working advisory groups (see Supplementary Figure S10).

The organizations established to integrate climate change adaptation measures into Myanmar's development plans and policies are described in Supplementary Table S1. To ensure Myanmar fulfills its obligations within the Paris Agreement's framework for increased transparency, a monitoring, reporting, and verification (MRV) system was established. To better comprehend top-level data and information systems required to establish the national MRV systems for complying with the standards of the enhanced transparency framework, MRV intends to develop institutional monitoring and reporting frameworks. For an MRV system of the Nationally Determined Contribution (NDC), MONREC is strengthening institutional and human capacity. Institutional coordination and multi-stakeholder engagement frameworks were established for the implementation of climate-smart actions in agricultural, fisheries, and livestock sectors [4,5]. However, Myanmar still requires financial assistance from the international community to achieve all of its NDC conditional goals, implement mitigation and adaptation strategies, and pursue the goal of promoting green recovery beyond COVID-19 [5].

### 5.2. Key Stakeholders and Adaptation Priorities

To efficiently implement climate change mitigation and adaptation actions in the agriculture, fishery, and livestock sectors, institutional coordination and a multi-stakeholder engagement framework have been established. Table 3 depicts important stakeholders by organizational type. Six prioritized sectors related to adaptation, mitigation, and cross-cutting challenges are described in the MCCS and Master Plan to accomplish two goals. The first involves making vulnerable communities and sectors more adaptable so they can resist the impact of climate change, and the second entails creating and maximizing opportunities for potential sectors to pursue low-carbon development pathways to ensure that households and all economic sectors will benefit from this development.

**Table 3.** Key players and stakeholders for climate change adaptation in Myanmar.

| Government | Private Sectors/Institutes | Community |
|---|---|---|
| <ul><li>Department of Meteorology and Hydrology (DMH)</li><li>Ministry of Agriculture, Livestock, and Irrigation (MOALI)</li><li>Environmental Conservation Department (ECD)</li><li>Forest Department</li><li>Department of Rural Development (DRD)</li><li>Irrigation and Water Utilization Management Department</li><li>Department of Disaster Management</li><li>Agricultural University, Institutes, and Research Centers</li></ul> | <ul><li>Local non-governmental organizations (NGOs)</li><li>Community-based organizations (CBOs)</li><li>Civil society organizations (CSOs)</li><li>Other private sectors (i.e., UMFCCI [1], Myanmar Climate Change Alliance)</li></ul> | <ul><li>Community-based associations and groups</li><li>Farmer's union and association</li><li>Youth-led voluntary group</li><li>Climate change activists</li></ul> |

[1] Union of Myanmar Federation of Chambers of Commerce and Industry.

According to the MCCS, adaptation, and mitigation concerning food security include climate-smart agriculture, fisheries, and livestock. The adaptation mechanism's goal is to encourage climate-resilient productivity and climate-smart responses in the agriculture, fisheries, and livestock sectors to support food and nutrition security while also encouraging resource-efficient and low-carbon practices that might improve the development of new markets and products. The MCCS strategy encourages a highly productive and competitive agriculture sector and aims to create a climate-resilient food-, water-, and nutrition-secure Myanmar by 2030 [4,5].

The climate change action plans have been incorporated into the pertinent policies, planning, and budgeting processes of the agriculture, fishery, and livestock sectors to promote climate-resilient productivity and climate-smart actions. Additionally, with the aid of local and international financing, many sectors have embraced adaptation techniques and technology that are both environmentally responsible and climate-resilient. Through livelihood diversification, climate-smart agriculture (CSA), climate and weather monitoring, and knowledge-sharing interventions, MOALI has collaborated with multiple stakeholders and organizations to advance climate-smart adaptation practices in the agriculture, fisheries, and livestock sectors [4,5].

### 6. Climate Change Adaptation Measures of Farm Households in Myanmar

The main objective of climate change adaptation is to reduce vulnerability and enhance resilience to climate change. Appendix A (Table A1) summarizes significant farm-level adaptation measures in Myanmar based on previous research findings and literature surveys. As stated earlier, farming systems differ in each agroecological zone in Myanmar. For example, farming systems in the lower parts of Myanmar, Mon State, and Tanintaryi Region are dominated by perennial crops such as rubber and palm oil, but the central

dry zone employs a mixed cropping pattern dominated by annual crops. There are also differences in the adoption of climate change adaptation measures by farm households in different agroecological zones. Some climate-smart agriculture practices and technologies are used in various agroecological zones, while others are used in relation to crop types and weather patterns. Traditional climate change adaptation practices have also been observed to lessen the effects of climate change on crop productivity, and there has been a gradual transition of traditional adaptation methods to recommended agricultural practices in Myanmar over the past year [33].

Socioeconomic and institutional factors, particularly at the farm level, influence the adoption of climate change adaptation methods [33,34]. Some measures will necessitate significant investment from the government, private sector, and individuals. Because modifying the existing agricultural system incurs some costs related to crop management and labor, the necessity to promote and scale up climate change adaptation measures for farm households in Myanmar is highlighted and suggested in Appendix A (Table A1).

## 7. Challenges and Constraints for Addressing Vulnerability and Increasing Resilience

In Myanmar, there is a growing assessment of farm-level adaptation constraints, farmer perceptions about climate change, and farm household adoption of adaptive strategies. Table 4 shows the primary constraints to the adoption of climate change adaptation methods, as well as the institutional challenges associated with the implementation of climate change adaptation measures, particularly in the agriculture sector.

**Table 4.** Constraints and challenges for addressing vulnerability and climate adaptation [1,2].

| Farm-Level Constraints | Institutional Challenges | Remarks |
|---|---|---|
| - Lack of credit<br>- High-interest, unregulated informal lenders<br>- Not having any credit collateral | - Poor credit scheme<br>- Unable to provide sufficient loans for all farmers in need | - Lack of understanding of the cropping cycle by private companies and banks for agricultural loans |
| - A shortage of skilled farm workers<br>- Majority of small landholding farmers and landlessness | - Administrative burden<br>- Civil unrest and war<br>- Land dispute | - Lack of technical knowledge and capacity<br>- Civil unrest and land confiscations |
| - Lack of agricultural inputs | - Low budget for climate actions (both mitigation and adaptation) | - Lack of awareness about climate change adaptation |
| - Low fertility of the soil<br>- High input prices | - Poor public extension service (i.e., encourage farmers to adopt advanced fertility technology for improving soil fertility in the long term) | - Lack of knowledge and awareness of soil structure maintenance<br>- Farm input prices increase along with a high inflation rate |
| - Lack of farm equipment and poorly mechanized<br>- Lack of draught animals for basic farming purposes | - Limited machinery rental services provided by the Agriculture Machinery Department and Myanmar Economic Holdings Company | - Due to the lack of collateral, it is difficult to apply hire-purchase arrangements offered by private banks and other commercial financial institutions in partnership with machinery retailers. |
| - Lack of access to sufficient water | - Limited infrastructure development<br>- Lack of law enforcement | - Monopoly and lack of proper maintenance on irrigation dams/weirs |
| - Food insecurity and malnutrition | - Challenges along with food security, poverty, hunger, and malnutrition | - Lack of awareness and capacity |

[1] [33]; [2] [34].

## 8. Discussion of Constraints to Effective Adaptation

In this section, we discuss constraints to adaptation, first at the farm level and then at the institutional level. Unlike previous studies, this study identifies the main barriers to farm households' adoption of climate change adaptation measures in Myanmar. One of the

major constraints is farmers' lack of awareness and capacity (see Figure 1). Many small-scale farmers lack knowledge of climate-smart agricultural practices and technologies that can assist them in adapting to climate change. Farmers' ability to adapt to climate change is hampered by a lack of awareness and capacity, resulting in crop failure, lower yields, and increased vulnerability to extreme weather events. Farmers also require technical knowledge about sustainable farming practices such as water management, soil conservation, and crop diversification to adapt to climate change. However, many farmers lack these skills and technical knowledge, making it difficult for them to adapt their farming practices to the changing climate.

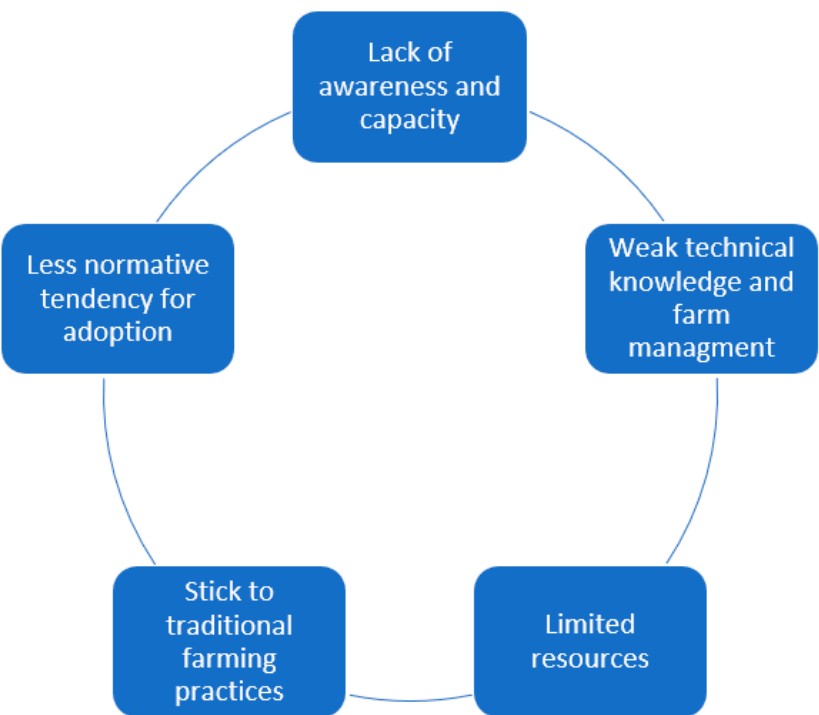

**Figure 1.** Major constraints in the adoption of farm-level adaptation measures. (Adapted from [8,24,28]).

Furthermore, many farmers do not have the financial means to invest in climate-smart technologies and practices. They may also lack access to credit, making it difficult for them to adopt new practices that require an initial investment. Farmers' ability to adapt to climate change is hindered by a lack of resources, making them more vulnerable to its effects. Another barrier to farm-level adaptation to climate change is a preference for traditional farming practices. Many farmers are resistant to change and prefer to stick to traditional methods and labor-intensive farming practices, even if they are not sustainable or appropriate for climate change [8,19,35,36]. Farmers' ability to adapt to climate change may be impeded by this resistance to change, resulting in lower productivity and increased vulnerability to extreme weather events. Finally, social norms and cultural barriers can also be a constraint to farm-level adaptation to climate change. Farmers may be hesitant to adopt climate-smart agricultural practices and technologies even if they have access to them due to the limited access to resources such as inputs, financing, or credits, as well as technology and extension services [22,33,37,38]. And, [38] demonstrated that farm households practicing young-seedling cultivation methods, such as the system of rice intensification (SRI), are hindered in adopting the entire set of techniques due to several limitations. Thus, this less-normative adoption tendency can impede the widespread adoption of climate-smart practices, limiting their impact on farmers' ability to adapt to climate change.

Furthermore, there are major constraints to integrating climate change adaptation policies in Myanmar to improve agriculture–food resilience. Myanmar confronts a number

of obstacles in integrating climate policies (see Figure 2). One of the major constraints is the lack of institutional capacity. The institutional capacity of Myanmar to implement and enforce climate policies is limited, owing primarily to a lack of expertise and skilled personnel. In addition, the country lacks adequate infrastructure (in particular, unreliable electricity supply and reliable roads) and financial and technical resources (i.e., weak public financial management and underinvestment in education and capacity development), which impedes its ability to implement and enforce policies. Another constraint is a lack of coordination and cooperation among various ministries. In the absence of effective inter-ministerial coordination, policies may lack coherence, impeding the achievement of climate goals. Because of its limited financial resources, the country must rely heavily on external funding to implement and enforce climate policies.

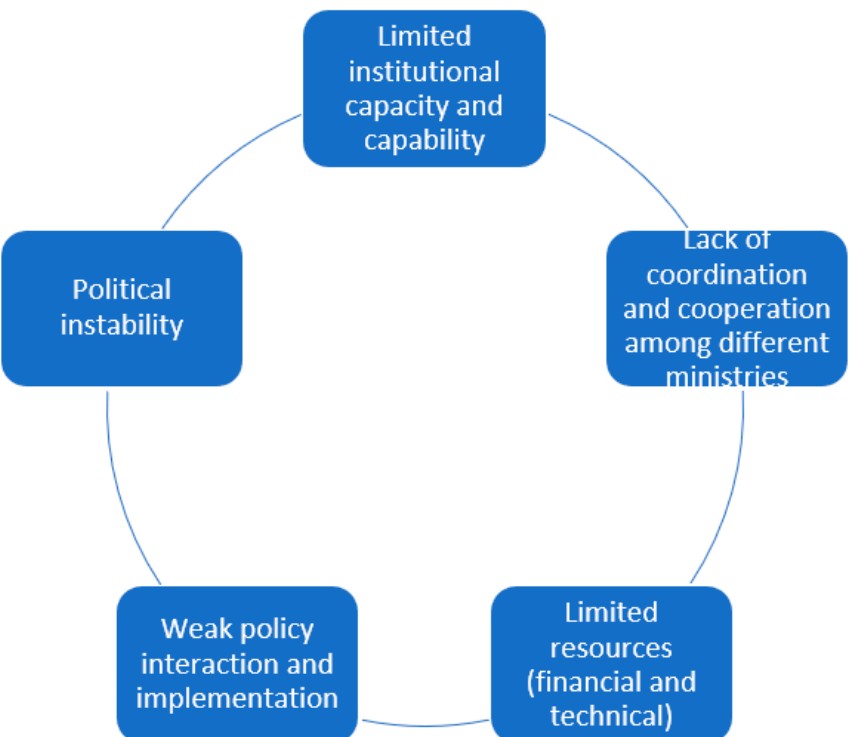

**Figure 2.** Major barriers to integrating climate policies into Myanmar's actions (adapted from [8,16,22,39]).

Another barrier to integrating climate policies in Myanmar is poor policy interaction and implementation. The country's policy implementation mechanism is weak, and it lacks the necessary legal and regulatory frameworks to support climate policy implementation. Finally, political instability is a significant impediment to Myanmar's implementation of climate policies. The country's history of political instability, combined with the ongoing conflict, makes maintaining continuity and consistency in policy implementation difficult. This instability may result in a lack of political will to prioritize climate policies, undermining planned climate actions and jeopardizing the country's ability to meet its climate goals.

## 9. Recommendations to Overcome Policy Coordination and Adaptation Constraints

Resolving the constraints identified in the previous section requires interventions or innovations in nine key areas.

(1) Investment: Promoting climate-smart agriculture (CSA) practices and adaptation strategies necessitates significant financial investment from both private and public sectors. Some CSA and adaptation measures require investment in research and extension services so that successive governments may provide updated climate adaptation solutions to farmers in places where farmers can extensively use these practices.

(2) Policy and Incentives: The Myanmar government should establish a more conducive environment for better farming system resilience and farm household adoption of CSA practices through policy and incentive initiatives. Farmers, for example, should be compensated for the value of their environmental contributions as well as the value of their physical production in environmentally appropriate methods.

(3) Private sector and other key stakeholders: Better regional and national planning and management mechanisms should be put in place to ensure improved networking and coordination between key stakeholders in Myanmar's government and private sectors. Climate change adaptation and mitigation actions, including the implementation of CSA initiatives, can be carried out more efficiently and effectively with these improved mechanisms. Implementing the climate-smart villages (CSV) initiative also requires extensive collaboration among individual farmers, government agencies, and other stakeholders.

(4) Capacity building training and extension services: It is also necessary to empower and strengthen the adaptive capacity of key stakeholders, policymakers, development planners, communities, and farmers. This can be accomplished by launching regular training programs, field-based extensions, services, and knowledge-sharing events in collaboration with both the public and private sectors. The farmer field school (FFS) is indeed a platform for farmers to learn innovative agricultural management practices and build new skills and knowledge.

(5) Improving climate change education and awareness: Improving education and awareness on climate change is crucial to building public support for climate action. There is still a need to identify effective education and awareness strategies and to evaluate their effectiveness in promoting climate action in Myanmar.

(6) Community-based resilience planning: Farmers' and communities' adaptive capacity and resilience can be strengthened through community-based resilience planning processes and farmers-to-farmers knowledge-sharing initiatives. It is also recommended that local civil society organizations (CSOs), community-based organizations (CBOs), and non-governmental organizations (NGOs) work more collaboratively and cooperatively on community-based resilience planning. Indigenous climate change adaption methods and community-based resource management practices should also be supported and encouraged.

(7) Early warning and early action system (EWEAS: Lack of knowledge on climate change and extreme weather events has negative effects on agricultural yield and farm income, as well as the loss of farm equipment and household possessions. The government of Myanmar needs to speed up the early warning and response system so that it can educate the public as soon as weather and climate conditions change and increase awareness of natural disasters and climate-change-related occurrences. Early weather and climate change information will, in the end, help farmers in taking early action and lessening agricultural losses.

(8) Enhancing climate data and monitoring: High-quality climate data are essential for effective climate change planning and decision making. Further research is needed to improve climate data and monitoring systems in Myanmar, including improving data collection and analysis combined with climate modeling tools.

(9) Development projects and programs: Scaling up and broadening the adoption of CSA initiatives is essential for farm households to become more climate-resilient. Along with recent political instability and civil turmoil, projects promoting climate resilience and development programs are being phased out. Therefore, more development programs and efforts need to be supported to enhance capabilities for climate change resilience and adaptation. Myanmar must act quickly and genuinely to implement its planned climate policies because the country is experiencing a worsening climate crisis.

## 10. Conclusions

Myanmar is one of the countries in the world that is most sensitive to climate change, and a climate crisis is unfolding. It will unavoidably have an impact on livelihoods based on crop, fishery, and livestock sectors. There is no single solution that can reverse the increasing effects of climate change in Myanmar. Although steps have been taken, there is still a need for increased and more extensive collaboration and coordination across government institutions and between public and private organizations. Therefore, it is crucial to create an environment that promotes coordination and collaboration among various stakeholders.

Climate change adaptation methods should be integrated into societal norms and fulfill community needs. Promoting CSA practices and CSV activities will eventually result in sustainable agriculture–food systems by enhancing agricultural productivity and mitigating the effects of climate change from agriculture sectors. To maximize the benefits of CSA and CSV initiatives in agri-food systems, government ministries and agencies such as the MONREC, MOALI, and NECCCCC should foster collaboration among various stakeholders.

Given the increasing occurrence of climate change in Myanmar, this study makes several recommendations for the public sector and stakeholders from private and civil society organizations. It is becoming increasingly important to strengthen the adaptive capacity and resilience of various stakeholders and farm households in Myanmar. However, the COVID-19 pandemic, as well as continuous civil turmoil and political crisis, has made addressing climate challenges more complicated. Myanmar must take immediate and concerted action to address the mounting impacts of climate change on agriculture and food systems. Priority should be given to facilitating the scaling up of climate-smart interventions. In addition, rules and policies must be helpful rather than restrictive so that Myanmar's climate change adaptation actions can be taken effectively and promptly. Therefore, to lessen the negative effects of climate change on agri-food systems in Myanmar, the government should support more initiatives and projects that increase the adaptive capacity of stakeholders and individuals, and support climate-smart agriculture initiatives, as well as implement community-based resilience planning throughout Myanmar.

**Supplementary Materials:** The following supporting information can be downloaded at: https://www.mdpi.com/article/10.3390/cli11060124/s1.

**Author Contributions:** Conceptualization, A.T.O. and D.B.; methodology, A.T.O.; formal analysis, A.T.O. and N.A.; resources, A.T.O. and N.A; writing—original draft preparation, A.T.O. and N.A.; writing—review and editing, A.T.O., D.B. and N.A.; visualization, A.T.O. and N.A.; supervision, D.B. All authors have read and agreed to the published version of the manuscript.

**Funding:** This research received no external funding.

**Institutional Review Board Statement:** Not applicable.

**Informed Consent Statement:** Not applicable.

**Data Availability Statement:** The data that has been used in confidential.

**Acknowledgments:** The authors would like to thank the International Food Policy Research Institute (IFPRI) and Bart Minten, senior research fellow/program leader of the IFPRI's Myanmar Strategy Support Program, for their guidance in writing this manuscript.

**Conflicts of Interest:** The authors declare no conflict of interest.

# Appendix A

**Table A1.** Farm-level climate change adaptation measures categorized by type of adaptation strategies.

| Farming Systems and Major Crop Production | Indigenous Adaptation Measures | Recommended or Introduced Adaptation Measures | Agro-Ecological Zone | Remarks |
|---|---|---|---|---|
| Mix cropping and diversification of farming systems as an adaptation to climate change for crops including rice, peas and beans, sesame, groundnut, maize, and other cash crops such as onion, melon, chili, etc. | Crop- and livestock-related strategies (e.g., sowing seeds from neighboring farmers, livestock breed from friends, or nearby villages) Double cropping (summer paddy program on irrigated paddy land) | Recommended strategies by the Department of Agriculture (DOA) (i.e., good agricultural practices (GAP)) Climate-change-related changes to cropping systems (stress-resistant varieties, adjustment to farming practices) Hybrid rice production technology Good agriculture practice (GAP) | Mostly in central dry zone (Sagaing, Mandalay, Magwe Region) | Need additional investment in research and extension services to achieve higher adoption of these adaptation technologies |
| Adaptation through farm management practices for rice, peas and beans, groundnut and sunflower, etc. | Adjusting planting time Cultural-related strategies such as fumigation, cleaning bushes, hand weeding, etc. | Adjusting cultivation methods (i.e., adjusting sowing dates with broadcasting methods, different forms of seedling methods, and transplanting) Utilization of fertilizers, chemical herbicides, and weedicides | Mostly in CDZ (Sagaing, Mandalay, Magwe Regions), Bago and Yangon Regions | Unsystematic application of chemical fertilizers and pesticides lead to soil structural deterioration and harm to both human and ecosystems Need additional capacity-building training for private sectors, farm households, and public sectors |
| Crop-related adaptation strategies | Crop diversification | Crop diversification and rotation (i.e., a mix of crop types/varieties and crop rotation techniques) to improve farmer's income and livelihoods | Mostly in central dry zone (Sagaing, Mandalay, Magwe Region) | Need additional research on which crops should be rotated and diversified |
| Soil and water management practices | Soil-related strategies such as manure application, cow-dung application, soil tillage practices, shallow plowing, etc.) | Advanced soil and plant management (organic manure application, plant pest protection, mulching, weeding techniques introduced by government departments, deep plowing with machines, etc.) | Mostly in central dry zone (Sagaing, Mandalay, Magwe Region), Bago, and Ayeyarwady Region | Need to strengthen the adaptive capacity of stakeholders from both private and public sectors as well as farm households |
| | Banding practices | Improved variety application (i.e., drought-resistant varieties, high-yielding varieties, short-duration varieties) | Ayeyarwady and Bago Region | Department of Agriculture and Research (DAR) (DAR developed eight varieties of deep-water rice and one submergence-tolerant rice variety for flood-prone areas), but there is a need for investment for further research and development |
| | Hand weeding and mulching practices | Combination of improved variety and recommended agricultural strategies | CDZ, Bago, and Ayeyarwady Regions | A combination of variety selection and good agricultural practices produces better outcomes on crop production than one farm management practice |
| Agroforestry production systems | Production of rice and annual crops Agroforestry practices Mangrove forestation | Introduction of cash crops, fruit trees/intensification of perennial crop production (i.e., palm oil and rubber) in Tanintharyi Region and Mon State | Coastline areas, particularly in Rakhine and Ayeyarwady Region Chin, Kayin, Kachi, n and Shan Tanintaryi and Mon states | Need additional investment in processing technologies Mangrove forests and a number of agroforestry practices to retard flooding, tidal surge, and further saltwater intrusion to fish ponds, and rice fields |
| Crop management system | Conventional seedling practices | Brine seeds or soak rice seeds in salt water before planting to resist saltwater intrusion problems System of rice intensification (SRI) | Mostly at coastline areas such as Ayeyarwady Mostly at Bago, Mandalay, and Sagaing | DAR developed eight varieties of deep-water rice and one submergence-tolerant rice variety for flood-prone areas |

**Table A1.** *Cont.*

| Farming Systems and Major Crop Production | Indigenous Adaptation Measures | Recommended or Introduced Adaptation Measures | Agro-Ecological Zone | Remarks |
|---|---|---|---|---|
| Other technology as adaptation Agricultural development for food security, and poverty reduction | Conventional crop production with a heavy amount of fertilization application Rice-based farming systems Cultivation of rain-fed crops Production of vegetables such as home gardening and household consumption | Climate-smart agriculture (CSA) would contribute to regional food security and environmental protection. Community-level nutrition education and awareness building | In several parts of the country | Need to scale up these good practices throughout Myanmar |
| | | Climate-Smart and Nutrition-Smart Villages (CSVs) | In several parts of the country | Need to scale up these good practices throughout Myanmar |
| | | Organic farming technology | In several parts of the country | Need to scale up these good practices throughout Myanmar |
| | | Biochar technology and other organic compost-making practices | In several parts of the country | Need to scale up these good practices throughout Myanmar |
| | | Hydroponic vegetable production | Mostly in central dry zone (Sagaing, Mandalay, Magwe Region) and Shan State | Need to scale up these good practices throughout Myanmar |
| | | Conservation agriculture technology | Mostly in central dry zone (Sagaing, Mandalay, Magwe Region), Shan, Kachin, and Chin State | Provide more public awareness training for conservation agriculture practices, particularly for Inlay lakes in Shan State and Indawgyi Lake in Kachin State |
| | Sun-dried practices | Paddy dryers (post-harvest management) | Ayeyarwady, Bago, Mandalay, and Sagaing | Need additional investment for good post-harvesting and food processing practices |
| Water management practices | Traditional rainwater harvesting practice | Weather-index-based crop insurance | Mostly in central dry zone and Nay Pyi Taw areas | Need additional investment from private sectors |
| | Submerged irrigation method | Alternative wetting and drying irrigation | Sagaing, Mandalay, and Bago Region | Need additional investment for drainage water management system |
| | Rainwater irrigation system | Improved water management and Irrigation practices through building water-resilient infrastructure | Mandalay, Magwe, Sagaing, Bago, and Yangon Regions | Need additional investment for irrigation and drainage systems |

Adapted from [8,18,22,24,33].

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
