# Peer review of "Climate Change Adaptation and the Agriculture–Food System in Myanmar"

_climate, doi:10.3390/cli11060124_

Round 1

Reviewer 1 Report

This article examined the historical climate information, and the anticipated effects of climate change on Myanmar's agricultural sectors. In-depth literature reviews and summaries of Myanmar's climate change adaptation efforts are included in the report, along with recommendations for targeted, locally appropriate actions to strengthen the country's agricultural sector's resilience

It can only publish in the journal after careful considering the given points. 

My main concern is that the article is submitted to the climate journal, and this journal is international. I am wondering that the focus of topic is narrow. However, if authors will include the literature from various countries, it will be beneficial for global audience. Therefore, I highly recommend to add at least one paragraph at the start of introduction where authors can highlight the issue of climate change worldwide as climate change is a global issue. I highly recommend to incorporate following studies in the paragraph.   

[1] Spatiotemporal Analysis of Meteorological and Hydrological Droughts and Their Propagations. Water 2021

Please add main research questions in the introduction. 

Please write the section of recommendations after the section of conclusion.

Author Response

Dear Reviewer (s),

We would like to express our sincere appreciation for your insightful comments. Please find attached our response to your constructive feedback on the paper.

"I highly recommend to add at least one paragraph at the start of introduction where authors can highlight the issue of climate change worldwide as climate change is a global issue"...

Thank you. We have now resolved this issue and added a paragraph about climate change from a global perspective, as well as the significance of the study on a regional level.

"Please add main research questions in the introduction". 

Thank you for your input. We have now included the review paper's key research questions and why this study is important for understanding the effects of climate change on agriculture.

Please write the section of recommendations after the section of conclusion.

Thanks. As suggested, this has been revised. A supplementary file with figures and tables is also attached for a better understanding of the impact of climate change on the agro-food system. Additionally, our co-author, native English speaker Prof. Duncan, has thoroughly reviewed the document. We believe that this revised edition meets your standards for publication in the climate journal. Thank you for your time and consideration.

Reviewer 2 Report

While the paper is written nicely, the paper has a logical structure, my major comment is that I don't see where the originality of this research is. The paper describes geographical location and how this explains climate variability, outlines government climate initiatives, challenges face by farmers and adaptation strategies used. The paper is more like a good descriptive study but there is no originality to it. At least the authors never made it clear what the original contribution of this paper is. 

Author Response

Dear Reviewer (s),

We would like to express our sincere appreciation for your insightful comments. Please find attached our response to your constructive feedback on the paper.

"The paper is more like a good descriptive study but there is no originality to it. At least the authors never made it clear what the original contribution of this paper is...."

We have now thoroughly improved our paper in response to your suggestions/comments. The main research questions have been added, along with explanations of why this study is so unique and important for agriculture-food systems in Myanmar and around the world. For example, unlike previous research, this study identified farm-level adaptation constraints and barriers to climate change adaptation policy and interaction. A supplementary file with additional information and data on climate change has also been added. We hope that this revised edition will provide readers with more thoughtful insights into climate change adaptation measures for improved agro-food sector resilience. Additionally, our co-author, native English speaker Prof. Duncan, has thoroughly reviewed the document. We believe that this revised edition meets your standards for publication in the climate journal. Thank you for your time and consideration.

Reviewer 3 Report

This manuscript is a detailed summary of the geographical features of Myanmar, which are important for agriculture, on the one hand, and the predicted climate change, which may change these characteristics, on the other hand. It also lists the factors that potentially endanger different types of agricultural production sites. This is followed by a draft of the responses to the challenges of climate change, as well as a description of the institutions dealing with it.

The work is well compiled and edited, a perfect report on the country's agriculturally relevant situation, changes, and different strategies.

I miss only one thing, namely that it does not contain new scientific results. Of course, it can be of some interest even without it, but I don't think this is the forum where it should be published.

For this reason, I do not recommend for publication, but if the editor decides to do so, it can of course be published.

Author Response

Dear Reviewer (s), 

We would like to express our sincere appreciation for your insightful comments. Please find attached our response to your constructive feedback on the paper.  

In response to your comments/suggestions, this paper has been thoroughly revised and improved. We have now included the main research questions as well as why this study is so unique and important in light of climate change and its effects on agriculture and food systems. Unlike previous studies, this paper has highlighted the importance of climate change adaptation for agro-food systems, identified barriers to farm households adopting climate change adaptation measures, and identified constraints in the implementation of climate change adaptation policies. In addition, we have now included a supplementary file containing climate change data in figures as well as information about the current climate change adaptation system. We hope that this revised edition will provide readers with more thoughtful insights into climate change adaptation measures for improved agro-food sector resilience. Additionally, our co-author, native English speaker Prof. Duncan, has thoroughly reviewed the document. We believe that this revised edition meets your standards for publication in the climate journal. Thank you for your time and consideration.

Round 2

Reviewer 1 Report

Accept. Satidfied with revisions. 

Author Response

Dear Reviewer,

Thank you for reviewing our paper and providing valuable feedback. We are pleased to hear that you agree to publish our paper. Your constructive comments have been very helpful in improving the quality of our paper.

Thank you very much indeed.

Reviewer 2 Report

Unfortunately, even after the revisions, the authors have failed to produce a paper that has a substantial contribution to the body of the existing knowledge. I find this paper very descriptive in nature, with no substantial scientific contribution. First, the novelty (significance of this study) of this study is never mentioned until page 12 and even there, the authors highlight their contribution as follows: "Unlike previous studies, this study identifies the main barriers to farm households' adoption of climate change adaptation measures in Myanmar."  Yet, all of the author's discussion of the main barriers is based off of the previous studies and not from the work of their own, e.g. through interviews with the farmers to identify what challenges they're facing and what needs to be done to deal with those challenges. So, this paper is nothing but a summary of other researchers' work and as a result I'm not able to see how this study contributes to the existing research on farmers' vulnerabilities to climate change in Maynmar. 

Author Response

Dear Reviewer,
We'd like to thank you again for your insightful comments on our paper. Our paper is a review paper, and it provides more precise, regionally tailored recommendations for improving agriculture sector resilience through a systematic review of studies on climate change impacts, vulnerability, and adaptation measures. This review paper will help stakeholders from the government and non-government sectors, as well as researchers and policymakers, address climate change threats in Myanmar through agricultural adaptation. Table 1 (page 5) summarizes farmer perceptions based on an extensive review of available literature and publishing works, and Table 2 (page 7) provides a key summary of the country's major agro-climatic zones and vulnerability status. These have never been studied or published before. The authors also created and visualized Figures 1 and 2, and the key findings of these analyzes were briefly discussed in the discussion section. Annex table-A is also produced by the authors by summarising climate change adaptation measures available at farm level throughout a systemic review of the literatures/published works. As a result, we believe that this review paper has provided descriptive results from a literature review and discussed key barriers/constraints to the adoption of climate change adaptation planning at the farm and institutional levels. This can be useful for stakeholders from public and private organizations, as well as for future research into the barriers/challenges to the adoption/scaling up of climate change adaptation strategies in Myanmar and other developing country contexts. With these considerations in mind, our revised edition meets your criteria for publication at climate paper.

Thank you very much indeed. 

Reviewer 3 Report

The authors have significantly expanded and revised the manuscript. Many missing data were included, highlighting the data and processes that were not clearly formulated until now. I still do not see the scientific impact of the manuscript as significant, but it contains a lot of useful information and conclusions. Due to the above, the manuscript can be recommended for publication.

Author Response

(The authors gave the same response as above.)
